# Pronounced expression of extracellular matrix proteoglycans regulated by Wnt pathway underlies the parallel evolution of lip hypertrophy in East African cichlids

Nagatoshi Machii[1], Ryo Hatashima[1], Tatsuya Niwa[1,2], Hideki Taguchi[1,2], Ismael A Kimirei[3], Hillary DJ Mrosso[4], Mitsuto Aibara[1], Tatsuki Nagasawa[1], Masato Nikaido[1]*

[1]School of Life Science and Technology, Tokyo Institute of Technology, Tokyo, Japan; [2]Cell Biology Center, Institute of Innovative Research, Tokyo Institute of Technology, Yokohama, Japan; [3]Tanzania Fisheries Research Institute, Dar es Salaam, United Republic of Tanzania; [4]Tanzania Fisheries Research Institute (TAFIRI), Mwanza Fisheries Research Center, Mwanza, United Republic of Tanzania

*For correspondence:
mnikaido@bio.titech.ac.jp

Competing interest: The authors declare that no competing interests exist.

## eLife Assessment

Cichlid fishes have attracted attention from a wide range of biologists because of their extensive species diversification at the ecological and phenotypic levels. In this **important** study, the authors have partially revealed the mechanism behind lip thickening in cichlid fishes, which has evolved independently across three lakes in Africa. To explore this phenomenon, the authors used histological comparison, proteomics, and transcriptomics, all of which are well suited for their objectives. With **compelling** evidence, this contribution provides insights into parallel evolution in polygenic traits and holds significant value for the field.

**Abstract** Cichlid fishes inhabiting the East African Great Lakes, Victoria, Malawi, and Tanganyika, are textbook examples of parallel evolution, as they have acquired similar traits independently in each of the three lakes during the process of adaptive radiation. In particular, 'hypertrophied lip' has been highlighted as a prominent example of parallel evolution. However, the underlying molecular mechanisms remain poorly understood. In this study, we conducted an integrated comparative analysis between the hypertrophied and normal lips of cichlids across three lakes based on histology, proteomics, and transcriptomics. Histological and proteomic analyses revealed that the hypertrophied lips were characterized by enlargement of the proteoglycan-rich layer, in which versican and periostin proteins were abundant. Transcriptome analysis revealed that the expression of extracellular matrix-related genes, including collagens, glycoproteins, and proteoglycans, was higher in hypertrophied lips, regardless of their phylogenetic relationships. In addition, the genes in Wnt signaling pathway, which is involved in promoting proteoglycan expression, was highly expressed in both the juvenile and adult stages of hypertrophied lips. Our comprehensive analyses showed that hypertrophied lips of the three different phylogenetic origins can be explained by similar proteomic and transcriptomic profiles, which may provide important clues into the molecular mechanisms underlying phenotypic parallelisms in East African cichlids.

## Introduction

Parallel evolution is evidence of natural selection that drives adaptive solutions to the environment (*Losos, 2011*). Acquisition of similar traits in distantly related lineages can be observed in multiple lineages, such as the loss of eyes in cave fish, patterns of wing colors in Heliconius butterflies, and body armor of sticklebacks (*Krishnan and Rohner, 2017*; *Stern, 2013*). Since it is unlikely that such a complex phenotype was acquired repeatedly by neutral evolution alone, the same direction of natural selection among species is likely to drive the parallel phenotypic change (*Losos, 2011*; *Stern, 2013*). Therefore, research on parallel evolution enables us to compare the commonalities and differences in the mechanisms behind the acquisition of novel traits and elucidate how adaptation occurs in similar environments.

East African cichlids offer one of the best opportunities to examine the molecular basis of parallel evolution (*Kocher et al., 1993*; *Muschick et al., 2012*). There are hundreds of endemic cichlid species in the East African Great Lakes, Victoria, Malawi, and Tanganyika, as a result of explosive adaptive radiation (*Salzburger, 2018*; *Santos et al., 2023*). As a result of independent adaptations in each lake, parallel evolution is often observed among lakes, such as that of the nose snout (*Duenser et al., 2023*), jaw morphology, and body shape (*Kocher et al., 1993*; *Muschick et al., 2012*). To reveal their evolutionary mechanisms, genetic bases of the phenotypes have been investigated. For example, the parallel evolution of body stripes in cichlids of Lake Victoria and Lake Malawi was explained by relatively simple changes in pigmental cell arrangement caused by mutations in the cis-regulatory element of agouti-related peptide 2 (agrp2) (*Kratochwil et al., 2018*). Although this parallel evolution of body stripe patterns can be explained by a single gene, most traits are likely to be polygenic, hindering elucidation of the parallel evolution mechanism.

Cichlid fishes have acquired a huge diversity in their oral morphologies, reflecting ecological and trophic specializations (*Albertson and Kocher, 2006*; *Kocher, 2004*; *Lecaudey et al., 2021*; *Lecaudey et al., 2019*). Hypertrophied lips, one of the most prominent examples of oral diversity, have been independently acquired in cichlids inhabiting each of the East African Great Lakes, as well as in the Nicaraguan crater lakes and South American rivers (*Henning et al., 2017*; *Kautt et al., 2020*; *Masonick et al., 2023*; *Masonick et al., 2022*). Several hypotheses have been proposed for the adaptive roles of hypertrophied lips, such as for effectively sucking prey out from narrow crevices (*Lukas et al., 2015*), reducing mechanical stress during foraging in rocky habitats (*Fryer, 1972*; *Greenwood, 1974*), and facilitating prey detection by providing an enlarged area for taste receptors (*Arnegard and Snoeks, 2001*; *Oliver and Arnegard, 2010*). Thus, hypertrophied lips are considered an adaptive trait for a specific trophic niche, making them an ideal model for understanding the mechanism of parallel evolution.

The molecular mechanisms underlying hypertrophied lips have been investigated in various lineages of cichlids. For example, transcriptome comparisons between hypertrophied and normal lips of cichlids in Lake Tanganyika (*Colombo et al., 2013*) and South American crater lakes *Manousaki et al., 2013* have identified some differentially expressed genes (DEGs) as candidates for lip hypertrophy; however, their primary regulators remain to be elucidated. In addition, quantitative trait locus (QTL) mapping (*Henning et al., 2017*) and genome-wide association study *Masonick et al., 2023* have been conducted to elucidate the genomic loci responsible for hypertrophied lips in the cichlids of Lake Victoria and Lake Malawi. These genome-wide studies concluded that lip hypertrophy was a polygenic trait involving several candidate genes. However, the molecular mechanism underlying lip hypertrophy has not yet been elucidated.

In this study, we comprehensively compared the hypertrophied and normal lips of cichlids across all East African Great Lakes at various biological levels using histology, proteomics, and transcriptomics. As a result, we showed that a novel key pathway commonly involved in the formation of hypertrophied lips, providing insight into a better understanding of the molecular basis of a typical example of parallel evolution.

## Results

### Comparative histology of hypertrophied and normal lips among cichlids from the East African Great Lakes

We compared the histological structures of hypertrophied and normal lips among cichlids from Lake Victoria, Lake Malawi, and Lake Tanganyika. For an accurate comparison, we first observed serial sections of the lower lips with the jaws to determine the midline of the lips in each species, which was defined by the shape of the lower dentary bone (*Figure 1—figure supplement 1*). Based on this definition, we compared the distribution of collagens and proteoglycans, the main extracellular matrix (ECM) components, at the midline of the lower lips using Van Gieson (VG) stain (red) and Alcian blue (AB) stain (blue). Hypertrophied lips of cichlids from all lakes generally consisted of two mutually exclusive structures in the hypodermis: a collagen-rich layer at the base and a proteoglycan-rich layer at the tip. Moreover, comparison of hypertrophied and normal lips showed that the area of the proteoglycan-rich (AB-positive) layer was notably enlarged in hypertrophied lips, which was statistically significant in cichlids from Lake Victoria (*Figure 1*; *Figure 1—figure supplement 2*).

### Proteoglycans and their related proteins were abundant in the hypertrophied lips

To explore the proteins associated with lip hypertrophy, we used shotgun proteomics to compare the protein profiles of hypertrophied and normal lips in Lake Victoria cichlids. We detected ionic peptide fragments using LC–MS/MS in DIA/SWATH acquisition mode and analyzed them using DIA-NN software. As a result, we identified 2720 proteins in total from three types of lip samples: the tip of hypertrophied lips (proteoglycan-rich loose connective tissue), the base of hypertrophied lips (collagen-rich loose connective tissue), and normal lips. First, we observed few differences between the tip of hypertrophied lips and the base of the hypertrophied lips (*Figure 2—figure supplement 1*). On the other hand, we identified 138 differentially accumulated proteins between the tip of hypertrophied and normal lips (133 proteins upregulated and 5 proteins downregulated in the hypertrophied lip) under the criteria (p-value <0.05; log2 fold change >1) (*Figure 2A*). Compared to normal lips, tip of hypertrophied lips is abundant in proteoglycans and their related proteins such as versican a/b (vcan), brevican (bcan), periostin (postn), hyaluronan and proteoglycan link protein 1 (hapln1), and hyaluronan-binding protein 2 (habp2), which were highly accumulated in hypertrophied lips with statistical significance (*Figure 2B–D*). The results of the above proteomic analyses were consistent with the histological observation that proteoglycan-rich loose connective tissue was enlarged in hypertrophied lips compared to that in normal lips.

### Chondroitin sulfate proteoglycans such as vcan are the main component of hypertrophied lips

Next, we performed glycosaminoglycan digestion experiments to examine whether loose connective tissue was composed of the proteoglycans, which were detected as highly accumulated proteins in hypertrophied lips. Among the proteoglycans, we focused on vcan, which showed a clear difference in protein amount between hypertrophied and normal lips. Vcan is a high-molecular-weight core protein (approximately 370 kDa) that is glycosylated with chondroitin sulfate (*Iozzo and Schaefer, 2015*; *Wight et al., 2020*). Therefore, we stained sections of hypertrophied lips with AB stain after the treatment with chondroitinase, which specifically digests glycosylated chondroitin sulfate. As a result, AB staining in loose connective tissue became drastically weak in treated sections, which was common among lakes (*Figure 2E–G*). Our data suggested that the main components of enlarged loose connective tissue in hypertrophied lips are chondroitin sulfate proteoglycans such as vcan.

### Convergent expression in ECM-related genes

Given that proteomic analysis is not well suited for detecting high-turnover proteins (*Decaris et al., 2014*), we additionally conducted transcriptomic comparisons between hypertrophied and normal lips across Lake Victoria, Lake Malawi, and Lake Tanganyika. Principal component analysis (PCA) with all expressed genes clustered according to each lake, reflecting genetic closeness (*Figure 3A*). Next, we compared the gene expression of ECM-related genes, for which the proteomics profiles were clearly distinct between hypertrophied and normal lips. A PCA plot was drawn using only ECM-related genes

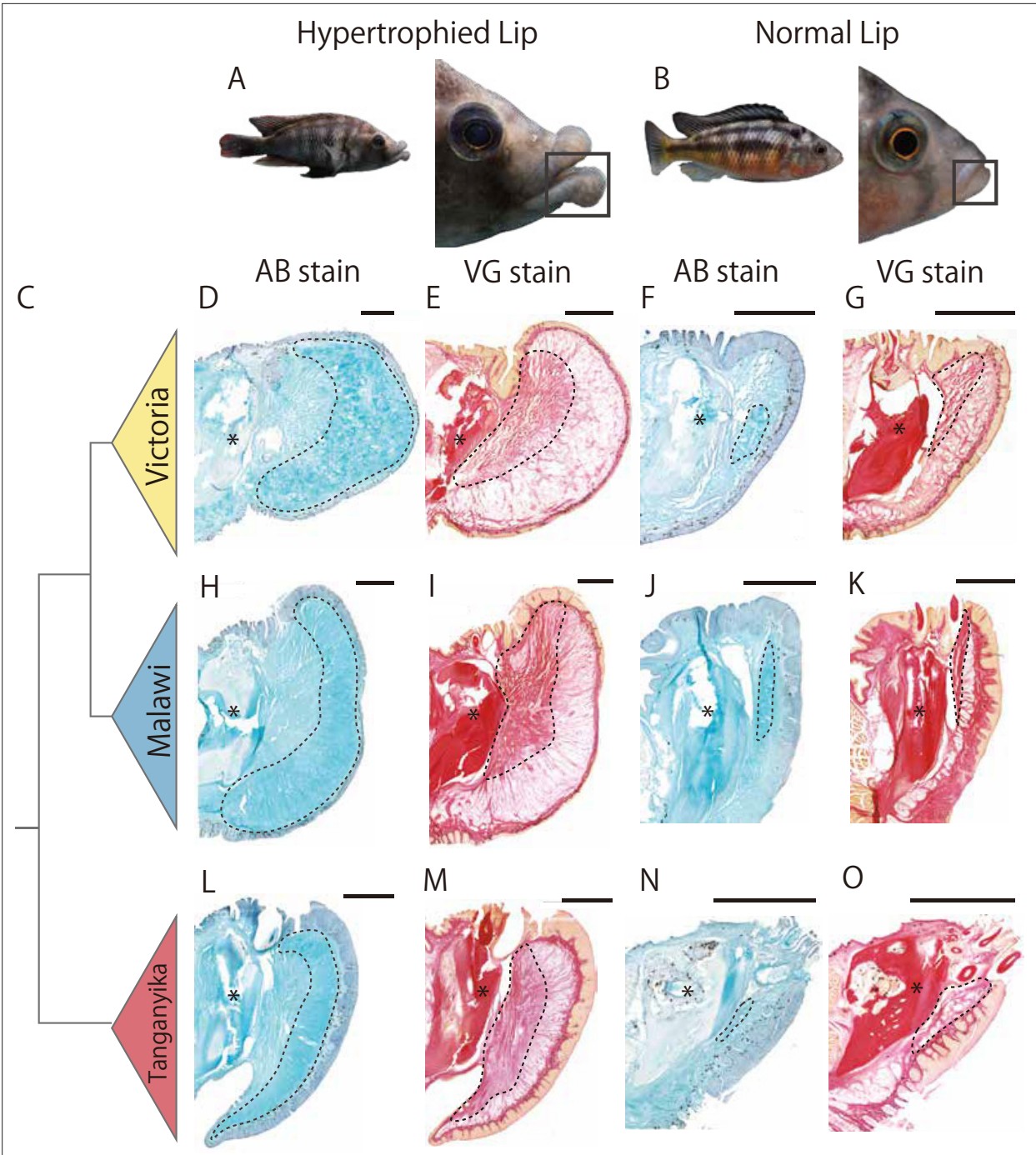

**Figure 1.** Histological comparisons of hypertrophied and normal lips of cichlids among lakes, focusing on proteoglycan-rich loose connective tissue. Examples of typical cichlids with hypertrophied lips (**A**, *Haplochromis chilotes*) and normal lips (**B**, *H. sauvagei*). The lower jaw and lip are highlighted by black squares. (**C**) Phylogenetic relationships among cichlids from Lake Victoria, Lake Malawi, and Lake Tanganyika. The midline of the lips, based on the lower dentary bone of the lower jaw (marked by asterisks), in cichlids from Lake Victoria (**D, E**, *H. chilotes* with hypertrophied lips; **F, G**, *H. sauvagei* with normal lips), Lake Malawi (**H, I**, *Placidochromis milomo* with hypertrophied lips; **J, K**, *Maylandia lombardoi* with normal lips), and Lake Tanganyika (**L, M**, *Lobochilotes labiatus* with hypertrophied lips; **N, O**, *Tropheus moorii* with normal lips) were sectioned in sagittal, and stained with AB: Alcian blue (**D, H, L, F, J**, and **N**), or VG: Van Gieson (**E, I, M, F, J**, and **N**). Each section is shown with the dorsal side up and the anterior side to the right. The area surrounded by dotted lines shows proteoglycan-rich tissue (stained dense blue by AB) or collagen-rich tissue (stained dense red by VG). Each species showed mutually exclusive staining by AB and VG, and AB-positive (proteoglycan-rich) tissues were commonly enlarged in hypertrophied lips among the cichlids from all three lakes with statistical significance. Scale bars are 1 mm.

The online version of this article includes the following figure supplement(s) for figure 1:

*Figure 1 continued on next page*

*Figure 1 continued*
**Figure supplement 1.** The midlines of the lips were defined by the shape of the lower dentary bone.
**Figure supplement 2.** The relative area of the proteoglycan-rich layer was significantly enlarged in the hypertrophied lips of Victoria cichlids.

in cichlids, which were identified based on a previous study (*Hynes and Naba, 2012*; *Figure 3B*). In contrast to the PCA plot of all expressed genes, that of ECM-related genes reflected differences of lip morphology (hypertrophied and normal), regardless of the genetic closeness. To evaluate the results of these PCAs, we calculated the average silhouette (AS) score, which assesses the cohesiveness of the cluster. In the PCA of all genes, the AS score of the cluster by lake (0.1601) was higher than that of the cluster by lip morphology (0.0849). In contrast, in the PCA of ECM-related genes, the AS score of the cluster by lip morphology (0.1798) was higher than that of the cluster by lake (0.0774). These results indicate that hypertrophied lips are characterized by the expression of ECM-related genes.

Next, we compared the expression of each ECM-related gene between hypertrophied and normal lips of cichlids from the three lakes (*Figure 3C, D*). As a result, several ECM-related genes showed higher expression in hypertrophied lips than in normal lips, such as type-I collagen fibrils (col1a1a, col1a1b, and col1a2), hyaluronan-binding proteoglycans (vcanb, hapln1a/b, and hapln3), periostins (postna/b), small leucine-rich proteoglycans (dcn, lum, and aspn), and microfibril-related genes (mfap2, emilin1b, emilin2a, emilin2b, and emilin3b). Consistent with the proteomics comparison, some proteoglycans and their related proteins (vcanb, hapln1a, and postna) showed significantly higher expression in hypertrophied lips than in normal lips. In conclusion, we observed convergent expression of ECM-related genes in hypertrophied lips of cichlids among the East African Great Lakes.

## High expression of ECM-related genes in the juvenile and adult stages of hypertrophied lips

Next, we performed transcriptome analysis by comparing juvenile and adult cichlids from Lake Victoria to examine the possible expression changes in ECM-related genes during development (*Figure 4A*). In the adult stage, the expression profiles of ECM-related genes between hypertrophied and normal lips showed obvious differences (low correlation value 0.75; orange square in *Figure 4B*), which was consistent with previous analyses (*Figure 3B*). In the juvenile stage, the difference was weaker than in adult stage (0.85; blue square in *Figure 4B*). Interestingly, the difference in the expression profiles between the juvenile and adult stages of hypertrophied lips (0.90; *Figure 4B*, black square) was lower than that of normal lips (0.83; *Figure 4B*, black dotted square). To examine the genes contributing to the differences in expression profiles, we performed hierarchical clustering analysis for the relative expression levels of each ECM-related gene (*Figure 4C, D*). The analyses consistently showed the higher expression of proteoglycans and their related genes (e.g. postna, vcanb, and hapln1b) in hypertrophied lips at both the juvenile and adult stages. These results indicated that the accumulation of proteoglycans, which leads to the lip hypertrophy, begins at the juvenile stage.

## The Wnt pathway-related genes are highly expressed in hypertrophied lips

To explore the molecular pathways involved in lip hypertrophy, we performed an enrichment analysis using DAVID (*Huang et al., 2009a*; *Huang et al., 2009b*) against DEGs obtained from the transcriptome. As a result of enrichment analysis, DEGs were categorized in the canonical and noncanonical Wnt signaling pathways, suggesting that high expression of genes in the Wnt signaling pathway is likely to be involved in the hypertrophied lips of juvenile and adult fish. Comparison of the developmental stages revealed differences in the expression patterns between the canonical and noncanonical Wnt pathways: the genes of the canonical Wnt pathway were highly expressed in both stages (*Figure 5C–I*), whereas those of the noncanonical pathway were highly expressed only in the adult stage (*Figure 5J–N*). We further compared the expression patterns of these Wnt signaling pathways in cichlids from Lake Malawi and Lake Tanganyika. Notably, the expression of genes for the Wnt pathway tend to be higher in the hypertrophied lip cichlids in Lake Malawi and Lake Tanganyika, although the differences were not as distinct as those found in cichlids from Lake Victoria (*Figure 6*). These results were consistent with those of previous studies that showed that the expression of ECM-related genes, such as vcan, was promoted by the Wnt signaling pathway (*Rahmani et al., 2005*). Continuous

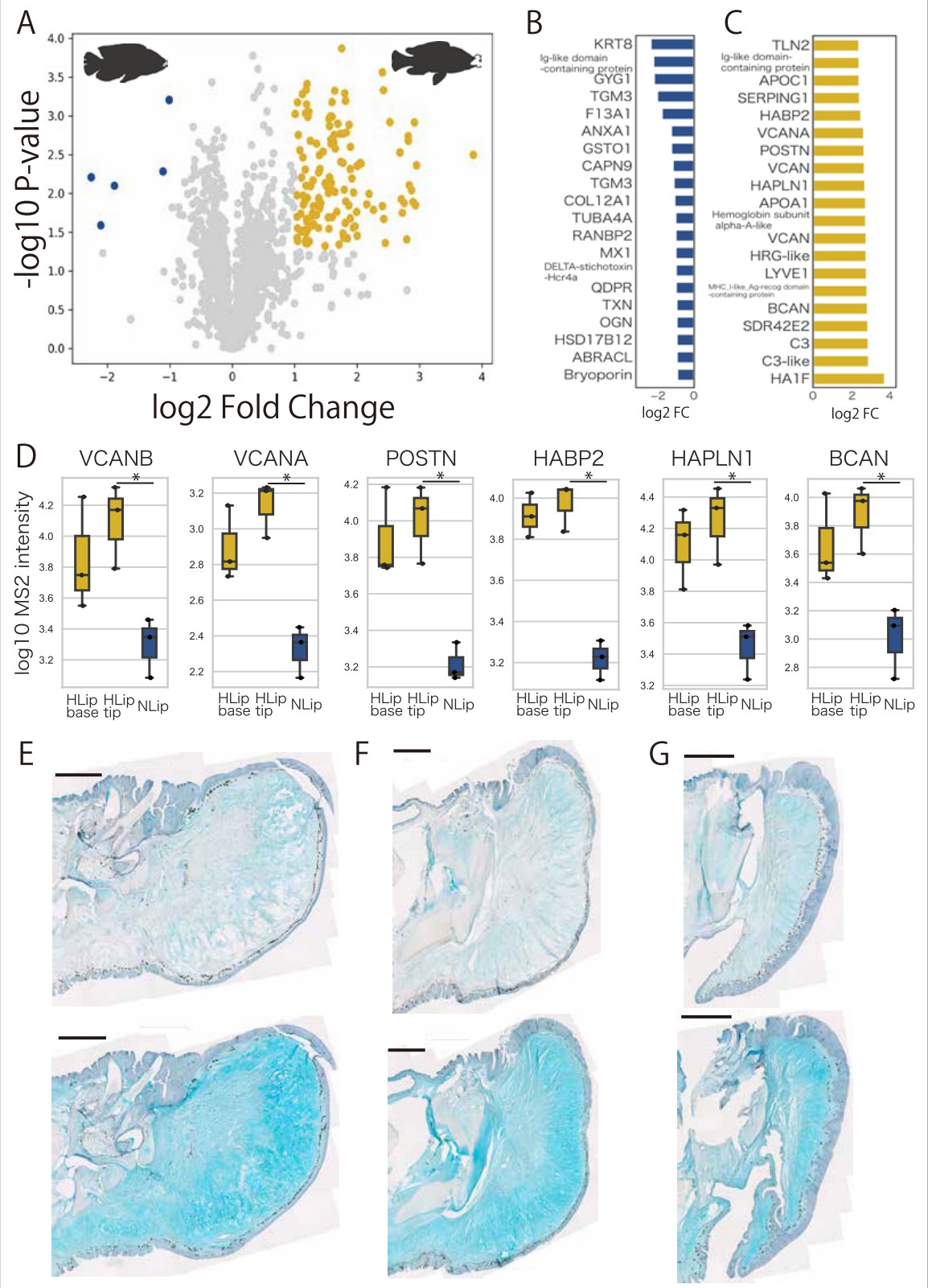

**Figure 2.** Chondroitin sulfate proteoglycans accumulate in hypertrophied lips. (**A**) Volcano plot of proteomics comparison between the tips of hypertrophied and normal lips. Statistical significance was examined using Welch's *t*-test. Yellow plots highlight highly accumulated proteins in hypertrophied lips, and blue plots highlight highly accumulated proteins in normal lips with the criteria of p-value <0.05 and log2 fold change >1. The top 20 proteins highly accumulated in normal (**B**) and hypertrophied lips (**C**) are shown along with log2 fold change relative to normal lips. (**D**) The

*Figure 2 continued on next page*

*Figure 2 continued*

relative amount of each proteoglycan-related protein. MS2 intensity indicates the intensity of peptides obtained from LC–MS/MS and DIA/SWATH analysis using DIA-NN software by comparing three parts of the lip: the base of hypertrophied lips (HLip), the tip of HLip, and normal lips (NLip). Welch's *t*-test was used to examine statistical significance (*p-value <0.05). Digestion of chondroitin sulfate by treatment with (upper panels) and without (lower panels) chondroitinase in hypertrophied lips of cichlids from Lake Victoria (**E**), Lake Malawi (**F**), and Lake Tanganyika (**G**). Each section is shown with the dorsal side up and the anterior side to the right. Scale bars are 1 mm.

The online version of this article includes the following figure supplement(s) for figure 2:

**Figure supplement 1.** Volcano plot of proteomics comparison between the tip of hypertrophied lips and the base of hypertrophied lips (**A**) and the base of hypertrophied and normal lips (**B**).

activation of the Wnt signaling pathway throughout all developmental stages may lead to lip hypertrophy via the deposition of ECM-related proteins.

## Discussion

In this study, we investigated the differences between hypertrophied and normal lips using histology, proteomics, and transcriptomics. Histological comparisons revealed that cichlids with hypertrophied lips from all three East African Great Lakes exhibited an enlargement of the proteoglycan-rich loose connective tissue. By integrating proteomics and transcriptomics, we identified several proteins that were significantly accumulated in the hypertrophied lips. Among them, proteoglycans and their related proteins such as vcan and postn, could be a primary factor that explains lip hypertrophy. Interestingly, these proteins are known to be involved in the formation of human keloids, a skin disease characterized by an overabundance of ECM in the dermis (*Deng et al., 2021*; *Naitoh et al., 2005*). It has been discussed that excessive deposition of vcan retains large amounts of water in the sulfate or carbonyl moieties, contributing to the increase in keloid volume (*Yagi et al., 2013*). In addition, postn is involved in fibrosis through its ability to interact with multiple ECM proteins. For example, upregulation of postn leads to strengthening of the ECM by crosslinking with collagens and excessive deposition of ECM proteins by differentiating quiescent fibroblasts into activated myofibroblasts (*Ashley et al., 2017*; *Yamaguchi et al., 2013*). In cichlid, vcan and postn may have a similar function, leading to hypertrophied loose connective tissue.

The Wnt signaling pathway was also elevated in both the adult and juvenile stages of hypertrophied lips. The Wnt signaling pathway is known to be a key regulator of the expression of ECM-related genes; for example, vcan is directly regulated by the Wnt/b-catenin and tcf4 signaling pathways, leading to vcan upregulation (*Rahmani et al., 2005*). Additionally, Wnt5a, a noncanonical pathway, plays a critical role in dermal condensation during mouse and sugar glider development (*Feigin et al., 2023*). Cellular-level localization of gene expression for the canonical and noncanonical Wnt pathway is required for future examination of the involvement of the Wnt signaling pathway in lip hypertrophy.

From a genomic perspective, several studies have investigated the genetic basis of hypertrophied lip cichlids (*Masonick et al., 2023*; *Nakamura et al., 2021*). Importantly, some Wnt pathway-related genes (tcf4 and daam2) and ECM-related genes (postna, col12a1a, and col12a1b) have been found to be under positive selection in cichlids with hypertrophied lips of Lake Victoria (see Table S3 in *Nakamura et al., 2021*). For future research, examining whether these genes are under selection in other lakes is crucial to understand the genetic mechanisms underlying the parallel evolution of hypertrophied lips.

Although the parallel evolution of hypertrophied lips in cichlids has received considerable attention in evolutionary biology, the commonality and diversity underlying their molecular basis remain poorly understood. This might be primarily due to the fact that many genetic factors are possibly involved in the dynamic organization of hypertrophied lips. We uncovered the apparent relationships between hypertrophied lips and the expression profiles of ECM proteins, in particularly proteoglycans. The trends for the overall expression of ECM-related genes were similar across hypertrophied lip species, but we rarely observed a specific gene that was commonly expressed at high or low levels in all three examples of hypertrophied lips across all East African Great Lakes. Furthermore, although we focused primarily on the relationship between the Wnt signaling pathway and lip hypertrophy, several other pathways may be involved in the development of hypertrophied lips. For example, our enrichment analysis in adult cichlids identified VEGF-related pathways, which could contribute to lip hypertrophy

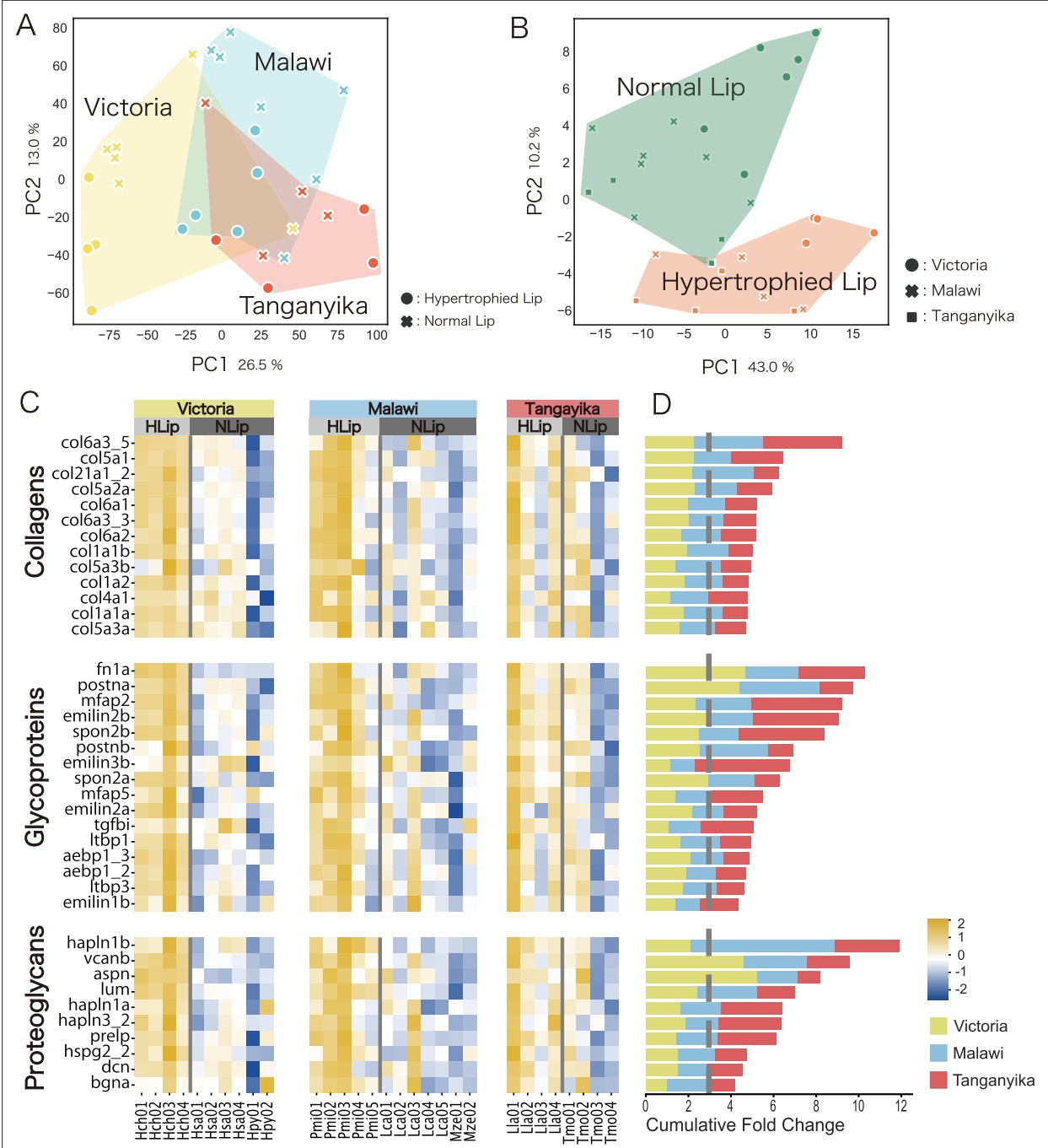

**Figure 3.** Hypertrophied lips of cichlids are characterized by higher expression of extracellular matrix (ECM)-related genes. Transcriptomic analysis using three species of Lake Victoria cichlids (*H. chilotes* (Hch) with hypertrophied lips; *H. sauvagei* (Hsa) and *H. pyrrhocephalus* (Hpy) with normal lips), three species of Lake Malawi cichlids (*Placidochromis milomo* (Pmi) with hypertrophied lips; *Maylandia zebra* (Mze) and *Labidochromis caeruleus* (Lca) with normal lips), and two species of Lake Tanganyika cichlids (*Lobochilotes labiatus* (Lla) with hypertrophied lips; *Tropheus moorii* (Tmo) with normal lips). Principal component analysis (PCA) plots of all expressed genes showed differences depending on the lakes (**A**), while those of ECM-related genes showed differences depending on lip morphology: hypertrophied or normal (**B**). (**C**) Heatmap of ECM-related gene expression between hypertrophied and normal lips. The expression rate (TPM) normalized by *Z* score of each gene is compared for Lake Victoria (yellow; Hch, Has, and Hpy), Lake Malawi (blue; Pmi, Lca, and Mze), and Lake Tanganyika (red; Lla and Tmo). The intensity of the relative expression rate is shown in the bottom right of the figure. (**D**) Cumulative fold change of the ECM-related genes comparing hypertrophied and normal lips. The gene symbol corresponds to (**C**). The color represents the lake information. The gray dotted line represents the value at which higher expression was observed in the hypertrophied lips.

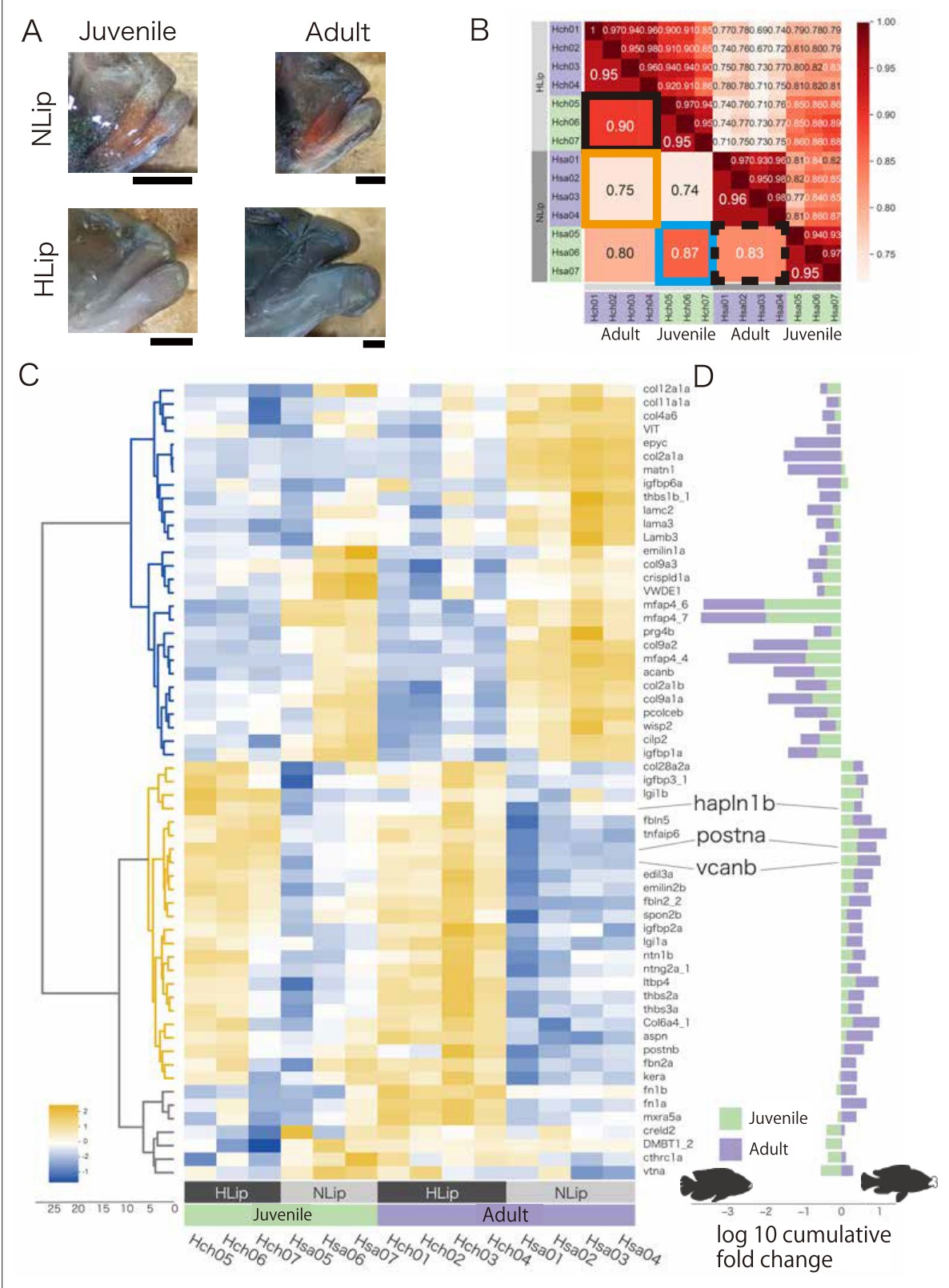

**Figure 4.** Comparison of juvenile and adult cichlids to examine gene expression changes during the development of hypertrophied lips. (**A**) The lips of the cichlids used in this analysis are shown with the dorsal side up and the anterior side to the right: the juvenile and adult stages of *Haplochromis chilotes* (Hch, representative of hypertrophied lips: HLip) and *H. sauvagei* (Hsa, representative of normal lips: NLip) from Lake Victoria. Hypertrophied lips at juvenile stage are not notably enlarged. Scale bars are 0.5 mm. (**B**) Spearman correlation matrix of the extracellular matrix (ECM)-related gene

*Figure 4 continued on next page*

*Figure 4 continued*

expression rate (TPM) normalized by *Z* score. The raw correlation values are indicated on the right side and the average correlation values within each stage and species are indicated on the left side. The orange frame highlights the correlations between the hypertrophied and normal lips at the adult stage, and the blue frame highlights those at the juvenile stage. The black frame highlights the correlations between juvenile and adult samples of each species. (**C**) Cluster heatmap of the ECM-related gene expression rate (TPM) normalized by *Z* score in each gene. Each gene is clustered by similarity of expression rates. The intensity of the relative expression rate is shown on the bottom left of the figure. (**D**) The log10 cumulative fold change of the ECM-related genes comparing hypertrophied and normal lips. The gene symbol corresponds to (**C**). The positive values indicate high expression in hypertrophied lips, whereas the negative values indicate low expression in hypertrophied lips. The colors of the bars indicate the stage of the samples.

by increasing vascularization and nutrient supply to the lip tissue. In addition, previous QTL analysis by *Henning et al., 2017* concluded that lip hypertrophy is likely influenced by numerous loci with small additive effects. These lines of data imply that although enlargement of proteoglycan-rich loose connective tissue is common in hypertrophied lips, the developmental pathways to accomplish this are diverse in each lake. The pathways responsible for lip hypertrophy therefore require further investigation. In conclusion, our approach of integrating histology, proteomics and transcriptomics revealed complex evolutionary pathways for hypertrophied lips in cichlids of the East African Great Lakes, providing an opportunity to further understand the molecular mechanism underlying the parallel evolution of polygenic traits, such as jaw morphology or body shape.

## Materials and methods
### Animals
Three species of Victoria cichlids (*Haplochromis chilotes* with hypertrophied lips; *H. sauvagei*, and *H. pyrrhocephalus* with normal lips), four species of Malawi cichlids (*Placidochromis milomo* with hypertrophied lips; *Maylandia lombardoi*, *M. zebra*, and *Labidochromis caeruleus* with normal lips), and two species of Taganyika cichlids (*Lobochilotes labiatus* with hypertrophied lips; *Tropheus moorii* with normal lips) were used in the experiments. *H. chilotes, H. sauvagei*, and *H. pyrrhocephalus* were collected from the Mwanza gulf of Lake Victoria during an exploration that was led by the Nikaido Laboratory in 2018. The other fish were obtained from a commercial supplier. Fish were maintained and bred at 27°C on a 12/12 hr light/dark cycle. One to 12 individuals were kept in a plastic tank (40 cm × 25 cm × 36 cm) and fed food pellets two to three times a day. The sample information is provided in *Supplementary file 1a*. All experiments were conducted in accordance with the Institutional Animal Experiment Committee of the Tokyo Institute of Technology.

### Histological observation
Cichlids were anesthetized with 0.02% ethyl p-aminobenzoate. The lower lips were then dissected, along with the lower jaw, and were fixed in 4% paraformaldehyde (Fujifilm-Wako, Chuo-ku, Osaka, Japan) in phosphate-buffered saline (PBS) at 4°C for 24 hr. Tissues were demineralized in 0.34 M ethylenediaminetetraacetic acid (EDTA, Fujifilm-Wako, Chuo-ku, Osaka, Japan) in 100 mM, pH 7.0 tris (hydroxymethyl) aminomethane (Tris, Fujifilm-Wako, Chuo-ku, Osaka, Japan) buffer at 4°C for 24 hr. After demineralization, tissues were treated with 20% sucrose/PBS at 4°C for 24 hr. Tissues were then embedded in Tissue-Tek O.C.T. compound (Sakura Finetek, Chuo-ku, Tokyo, Japan) and frozen with liquid nitrogen. Embedded blocks were sliced in the sagittal plane into 20 μm sections and placed on a coated glass slide (MAS-01, Matsunami, Bunkyo-ku, Tokyo, Japan). Sections were stored at −80°C until they were used for experiments. To observe the histological structure, hematoxylin–eosin (HE) staining, AB staining, and VG staining were conducted. Mayer's hematoxylin solution (Fujifilm-Wako, Chuo-ku, Osaka, Japan), Eosin Y solution (Fujifilm-Wako, Chuo-ku, Osaka, Japan), ALCIAN BLUE 8GX (Sigma-Aldrich, St. Louis, MO, USA), Van Gieson solution P (Fujifilm-Wako, Chuo-ku, Osaka, Japan), and Van Gieson solution F (Fujifilm-Wako, Chuo-ku, Osaka, Japan) were used for each staining. Experiments were performed according to the manufacturer's protocol and previous studies (*Arnegard and Snoeks, 2001*; *Oliver and Arnegard, 2010*). For AB staining, the pH was adjusted to 2.5 with 3% acetic acid (Fujifilm-Wako, Chuo-ku, Osaka, Japan). For VG staining, Van Gieson solution P and Van Gieson solution F were prepared at a ratio of 7.55:1. The stained area was calculated using ImageJ version 1.5.1 (*Schneider et al., 2012*) after trimming the image to include only the lips. The images were captured using a Zeiss Axioplan fluorescence microscope and a Zeiss Axiocam 503 color CCD

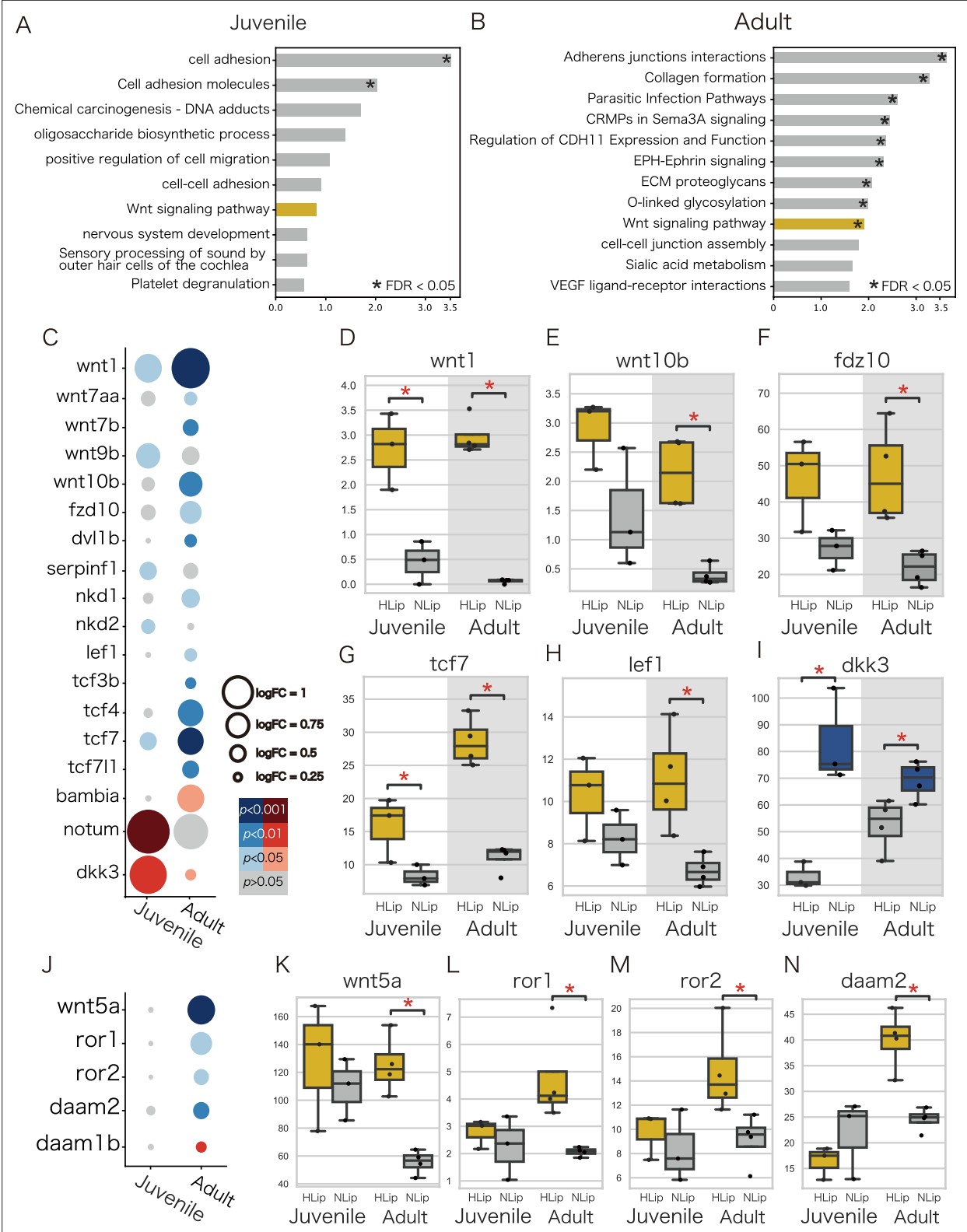

**Figure 5.** Wnt pathway-related genes are highly expressed in hypertrophied lips of juvenile and adult cichlids. (**A, B**) Enrichment analysis of highly expressed genes in the hypertrophied lips. Labels are according to the top annotation in DAVID annotation clusters. Asterisks indicate statistically significant annotation clusters (FDR <0.05). The Wnt-related annotations are highlighted in yellow. (**C–N**) Expression of the specific Wnt pathway-related genes. (**C, J**) The size of the dots represents the log fold change, and the color of the dots represents the p-value. Blue dots represent genes that are

*Figure 5 continued on next page*

*Figure 5 continued*

highly expressed in hypertrophied lips (HLip), and red dots represent genes that are highly expressed in normal lips (NLip). (**D–I, K–N**) Box plots show the expression rate (TPM) of Wnt pathway-related genes. Yellow boxes represent genes that are highly expressed in hypertrophied lips and blue boxes represent genes that are highly expressed in normal lips, respectively. Mann–Whitney *U*-test was used to examine the statistical significance (*p-value <0.05). (**C–I**) The canonical Wnt pathway-related genes and (**J–N**) the noncanonical Wnt pathway-related genes.

camera (Carl Zeiss, Oberkochen, Germany). Images were level-corrected, contrast adjusted, and tiled using Adobe Photoshop CC 2022.

## Chondroitinase digestion

Chondroitinase ABC from *Proteus vulgaris* (Sigma-Aldrich, St. Louis, MO, USA) was used for chondroitinase digestion. 2 mUnits/0.1% bovine serum albumin (BSA) solution in 20 mmol/l Tris-HCl pH 7.7 were prepared as the enzyme solution. In the experiment, the sections were first treated with 100 μl of 0.1% BSA solution for 5 min. Next, 100 μl of enzyme solution, or BSA solution for the control, was added. During the reaction, the sections were placed in a moist chamber. The enzymatic reaction was performed at 37°C for 1 hr. After the reaction, the samples were washed with water for 15 min, and then AB staining was examined.

## Identification of ECM-related genes in cichlids

To compare the transcriptome of cichlid lips, cichlid ECM-related genes were identified using the data from a previous study (**Hynes and Naba, 2012**). The amino acid sequences of human, mouse, and zebrafish ECM-related genes registered in Human GENCODE 41, Mouse GENCODE M30, Ensembl, and UniProtKB were used as queries, and those of *M. zebra* and *O. niloticus* were identified from their protein sequences registered in Ensembl and Uniprot using OrthoFinder2 v.2.5.4. The longest amino acid sequence of each isoform was selected to represent the gene. After Orthfinder2 analysis, ortholog correspondences were corrected manually, if necessary. A list of genes is provided in *Supplementary file 1b*.

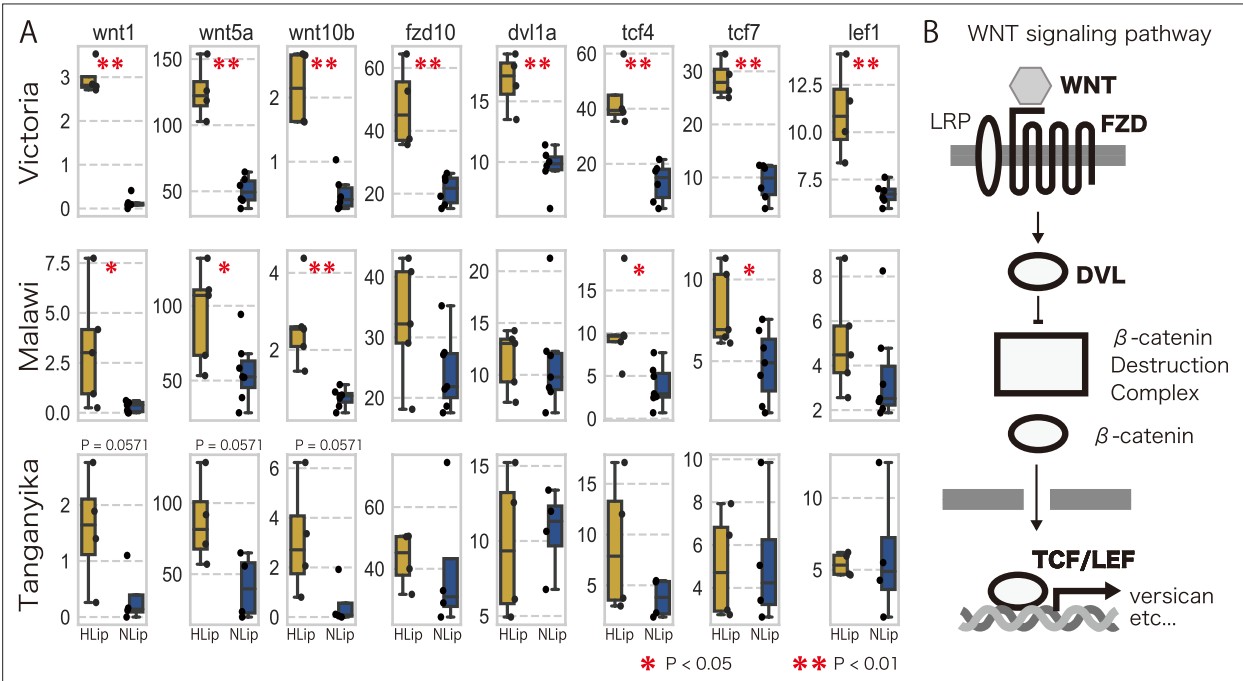

**Figure 6.** The trend of higher expression of Wnt-related genes in hypertrophied lips. (**A**) Box plots of expression rate (TPM) of Wnt-related genes of hypertrophied lips (HLip: yellow) and normal lips (NLip: blue). Mann–Whitney *U*-test was used to examine the statistical significance (*p-value <0.05, **p-value <0.01). (**B**) Illustration of the canonical Wnt pathway. Differentially expressed genes (DEGs) are marked in the bold.

## Sample preparation for proteomic analysis

Proteomic analysis was conducted on the lips of cichlids, *H. chilotes* (hypertrophied) and *H. sauvagei* (normal), from Lake Victoria. The lips of *H. chilotes* were dissected in two half anterior (tip) and half posterior (base), which are analyzed separately. For proteomics analysis of pellet fractions, dissected samples were homogenized with a pestle in 1% sodium dodecyl sulfate in PBS supplemented with a protease inhibitor cocktail (cOmpleteTM mini, EDTA free, Roche, Basel, Switzerland). The resultant lysate was centrifuged at $20,000 \times g$ for 15 min, and the supernatant was extracted. A fourfold volume of methanol, an equal volume of chloroform, and a threefold volume of distilled water were added to the supernatants, and the suspension was vortexed vigorously for 1 min after each addition. After vortexing, the supernatants were centrifuged at $15,000 \times g$ for 2 min and the upper water layer was removed. A threefold volume of methanol was added, and the samples were centrifuged at $15,000 \times g$ for 2 min. The supernatant was removed by aspiration and the pellet fraction was obtained.

Subsequently, sample preparation for LC–MS/MS was performed according to a previous study (*Masuda et al., 2008*), with some modifications. Proteins in the pellet fraction were solubilized in phase transfer surfactant (PTS) solution (12 mM sodium deoxycholate, 12 mM sodium lauryl sulfate, 100 mM Tris-HCl, pH 9.0), and the protein concentration was determined using a Pierce BCA Protein Assay Kit (Thermo Fisher Scientific, Waltham, MA). Then, 50 µg of protein in 50 µl of PTS was prepared for the following procedure. First, the proteins were reduced by treatment with 5 mM dithiothreitol at room temperature for 30 min, and then alkylated with 50 mM iodoacetamide in the dark at room temperature for 30 min. The reaction was stopped by a fivefold dilution with 50 mM ammonium bicarbonate. Denatured proteins in the solution were digested into peptide fragments in two steps: first, incubation with 0.5 µg of Trypsin/Lys-C Mix (Promega, Madison, WI) at room temperature for 3 hr, followed by incubation with 1.0 µg of Trypsin/Lys-C Mix at 37°C overnight. After digestion, an equal volume of ethyl acetate and 0.5% trifluoroacetic acid (final concentration) were added and the mixture was shaken vigorously for 2 min. The mixture was then centrifuged at $15,700 \times g$ for 2 min and the upper ethyl acetate layer was removed. The solvent was dried using a centrifugal evaporator. The residual pellet was redissolved in 600 µl of 0.1% TFA and 2% acetonitrile and then desalted using a handmade StageTip composed of an SDB-XC Empore Disk (3 M, Maplewood, MN, USA). The solution was applied to a StageTip equilibrated with 0.1% TFA and 2% acetonitrile, washed with 0.1% TFA and 2% acetonitrile, and eluted with 0.1% TFA and 80% acetonitrile. After the desalting, the solvent was dried using a centrifugal evaporator, and the residual peptides were redissolved in 120 µl of 0.1% TFA and 2% acetonitrile. The solution was centrifuged at $20,000 \times g$ for 5 min, and the supernatant was collected and subjected to LC–MS/MS measurement.

## LC–MS/MS measurement and data processing

LC–MS/MS measurements and SWATH-MS acquisition (*Gillet et al., 2012*) were performed using an Eksigent NanoLC Ultra and TripleTOF 4600 tandem-mass spectrometer or an Eksigent nanoLC 415 and TripleTOF 6600 mass spectrometer (AB Sciex, USA). The trap column used for nanoLC was a 5.0 mm × 0.3 mm ODS column (L-column2, CERI, Bunkyo-ku, Tokyo, Japan), and the separation column was a 12.5 cm × 75 µm capillary column packed with 3 µm C18- silica particles (Nikkyo Technos, Bunkyo-ku, Tokyo, Japan). The flow rate of nanoLC was 300 nl/min, and peptides were separated by applying a 10–40% linear acetonitrile gradient over 70 min in the presence of 0.1% formic acid. Data were acquired in DIA/SAWTH (*Gillet et al., 2012*) mode controlled by Analyst software (version 1.7, AB Sciex, USA) with the following settings: 15 Da fixed width window in 350–1100 *m/z* (50 cycles), 45 ms acquisition time, and 100–1600 fragment ion *m/z* range. The measurements were performed three times for each sample. Quantitative values were calculated by DIA-NN software (version 1.8.1) with default settings (*Demichev et al., 2020*). The library for DIA/SWATH quantification was generated using DIA-NN with the amino acid sequences of *M. zebra* total proteins obtained from UniProtKB. Our proteomics data was posted to jPOST (*Okuda et al., 2017*). The p-value and fold change were calculated by Python libraries (scipy.stats and pandas) using the proteins detected in all three measurements for three parts of the lips. Quantitative values were scaled by log10 before statistical tests (Welch's *t*-test, two sided).

## RNA-seq analysis

RNA was extracted from dissected lower lips using TRI Reagent (Molecular Research Center, Inc). Three species of Lake Victoria cichlids (*H. chilotes* [*n* = 4], *H. sauvagei* [*n* = 4], *H. pyrrhocephalus* [*n* = 2]), three species of Lake Malawi cichlids (*P. milomo* [*n* = 5], *Maylandia zebra* [*n* = 2], *L. caeruleus* [*n* = 5]), and two species of Lake Taganyika cichlids (*L. labiatus* [*n* = 4], *T. moorii* [*n* = 4]) were used for RNA-seq (*Supplementary file 1a*). The amount of sequence data ranged from 4.22 Gb (SAMD00762492) to 13.91 Gb (SAMD00024161). The extracted total RNA was sequenced at 100 bp paired-end reads on a Illumina NovaSeq 6000 by Macrogen Japan Corp, using a TruSeq stranded mRNA Library Prep Kit (Illumina). The raw FASTQ data were qualified using fastp version 0.20.1 (*Chen et al., 2018*) in default option. The high-quality reads were mapped to *M. zebra* genome (*Conte and Kocher, 2015*) using STAR version 2.7.10a (*Dobin et al., 2013*) and quantified using rsem-calculate-expression version 1.3.1 (*Li and Dewey, 2011*). PCA was conducted using Python scikit-learn package. Low-expression genes (average transcripts per kilobase million (TPM) below 5.0) were removed, and expression rates were normalized by $Z$ score. The resulting 12,231 genes were used for PCA analysis. For PCA of ECM-related genes, we used 186 genes as inputs, which were extracted in the same manner. DEGs were detected using the TCC package (*Sun et al., 2013*) with the TMM–edgeR–edgeR combination. Enrichment analysis was conducted using DAVID version 2023q3 (*Huang et al., 2009a*; *Huang et al., 2009b*) after the gene IDs of *M. zebra* were converted to human gene IDs using ensembl biomart. The annotation terms used for functional annotation clustering were collected from biological process (Gene Ontology), KEGG, and Reactome.

## Acknowledgements

This study was supported by JSPS KAKENHI (23KJ0951 to NM, 17H04606, 20KK0167 to MN) and JST SPRING (JPMJSP2106). Computations were partially performed on the NIG supercomputer at ROIS National Institute of Genetics. The authors thank the Open Research Facilities for Life Science and Technology, Tokyo Institute of Technology for LC–MS/MS analysis (for technical assistance). The authors thank the Hongoh laboratory members. Nagatoshi Machii thank my family, Nagashi Machii, Mitsue M, and Sawa M.

---

## Additional information

### Funding

| Funder | Grant reference number | Author |
|---|---|---|
| Japan Society for the Promotion of Science | 23KJ0951 | Nagatoshi Machii |
| Japan Society for the Promotion of Science | 17H04606 | Masato Nikaido |
| Japan Society for the Promotion of Science | 20KK0167 | Masato Nikaido |
| Japan Science and Technology Agency | JPMJSP2106 | Ryo Hatashima |

The funders had no role in study design, data collection, and interpretation, or the decision to submit the work for publication.

### Author contributions

Nagatoshi Machii, Conceptualization, Data curation, Formal analysis, Funding acquisition, Investigation, Visualization, Methodology, Writing - original draft, Writing - review and editing; Ryo Hatashima, Formal analysis, Investigation, Methodology; Tatsuya Niwa, Resources, Formal analysis, Investigation, Visualization, Methodology, Writing - review and editing; Hideki Taguchi, Resources, Methodology; Ismael A Kimirei, Hillary DJ Mrosso, Mitsuto Aibara, Resources; Tatsuki Nagasawa, Methodology, Writing - review and editing; Masato Nikaido, Conceptualization, Supervision, Funding acquisition, Visualization, Project administration, Writing - review and editing

### Author ORCIDs
Nagatoshi Machii ⓘ https://orcid.org/0009-0003-8703-339X
Ryo Hatashima ⓘ https://orcid.org/0000-0003-2734-1027
Hideki Taguchi ⓘ https://orcid.org/0000-0002-6612-9339
Mitsuto Aibara ⓘ https://orcid.org/0000-0001-8526-7449
Masato Nikaido ⓘ https://orcid.org/0000-0002-2430-6974

### Ethics
All experiments were conducted according to the Institutional Animal Experiment Committee of the Tokyo Institute of Technology. Reference number: 2023-032.

Reviewer #1 (Public review): https://doi.org/10.7554/eLife.99160.3.sa1
Author response https://doi.org/10.7554/eLife.99160.3.sa2

---

## Additional files

### Supplementary files
Supplementary file 1. Supplementary files for DEG analysis and Proteomics analysis in this study. (a) Basic information of the cichlids used in this study. (b) List of extracellular matrix (ECM)-related genes of cichlids identified in this study. The classification of ECM-related genes (the column of matrisome_annotation) follows a previous study (*Hynes and Naba, 2012*). (c) All differentially accumulated proteins detected by LC–MS/MS under the criteria of p-value <0.05, log2 fold change >1. (d) Results of enrichment analysis of highly expressed genes in hypertrophied lips of juvenile and adult cichlids. The annotation terms were clustered by DAVID annotation clustering. (f) All differentially expressed genes between hypertrophied and normal lips. We compared adult Victoria cichlids (*H. chilotes* with hypertrophied lips and *H. sauvagei* with normal lips), juvenile Victoria cichlids (*H. chilotes* and *H. sauvagei*), Malawi cichlids (*Placidochromis milomo* with hypertrophied lips and *Labidochromis caeruleus* with normal lips), and Taganyika cichlids (*Lobochilotes labiatus* with hypertrophied lips and *Tropheus moorii* with normal lips).

MDAR checklist

### Data availability
RNA-seq short reads and raw proteome data are stored at NCBI and jPOST (*Okuda et al., 2017*) (Accession: PRJDB3401, PRJDB17757, and JPST003042/PXD051571).

The following datasets were generated:

| Author(s) | Year | Dataset title | Dataset URL | Database and Identifier |
|---|---|---|---|---|
| Niwa T | 2025 | Proteomics data of cichlid hypertrophied and normal lips | https://repository.jpostdb.org/entry/JPST003042 | jPOST, JPST003042 |
| Hatashima R | 2025 | RNAseq of cichlid lip | https://www.ncbi.nlm.nih.gov/bioproject/PRJDB3401/ | NCBI BioProject, PRJDB3401 |
| Machii N | 2025 | RNAseq of cichlid lip | https://www.ncbi.nlm.nih.gov/bioproject/PRJDB17757/ | NCBI BioProject, PRJDB17757 |

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
