## [Editor Report · eLife Assessment]

Cichlid fishes have attracted attention from a wide range of biologists because of their extensive species diversification at the ecological and phenotypic levels. In this **important** study, the authors have partially revealed the mechanism behind lip thickening in cichlid fishes, which has evolved independently across three lakes in Africa. To explore this phenomenon, the authors used histological comparison, proteomics, and transcriptomics, all of which are well suited for their objectives. With **compelling** evidence, this contribution provides insights into parallel evolution in polygenic traits and holds significant value for the field.

---

## [Referee Report · Reviewer #1 (Public review)]

Summary:

Machii et al. reported a possible molecular mechanism underlying the parallel evolution of lip hypertrophy in African cichlids. The multifaceted approach taken in this manuscript is highly valued, as it uses histology, proteomics, and transcriptomics to reveal how phylogenetically distinct thick-lips have evolved in parallel. Findings from histology and proteomics connected to wnt signaling through the transcriptome are very exciting.

Strengths:

There is consistency between the results and it is possible to make a strong argument from the results.

Comments on revised version:

The issues I pointed out in the previous review have been carefully answered, and all issues have been addressed. The main points of the manuscript are clear, and the conclusions are easy to understand. The enlarged lips are a notable example of convergent evolution in African cichlids.

---

## [Author Response]

The following is the authors’ response to the original reviews.

**Public Reviews:**

**Reviewer 1:**
Weaknesses:The authors do not discuss based on genomic information; the genomes of the cichlids from the three lakes have been decoded and are therefore available. However, indeed, the species in Lake Tanganyika and Lake Malawi/Victoria are genetically distant from each other, so a comparative genome analysis would not have yielded the results presented here. I recommend adding such a discussion to the Discussion.

We appreciate your comment. We added the discussion regarding the genomic aspect of parallel evolution.

Line 386-393: “From a genomic perspective, several studies have investigated the genetic basis of hypertrophied lip cichlids (Masonick et al., 2023; Nakamura et al., 2021). Importantly, some Wnt pathway-related genes (tcf4 and daam2) and ECM-related genes (postna, col12a1a, and col12a1b) have been found to be under positive selection in cichlids with hypertrophied lips of Lake Victoria (see Nakamura et al., 2021 Table S3). For future research, examining whether these genes are under selection in other lakes is crucial to understand the genetic mechanisms underlying the parallel evolution of hypertrophied lips.”

Minor comments:Line 30, the Wnt  the genes in Wnt

We appreciate your comment. According to the comment, we corrected the sentence.

Line 30: “the Wnt signaling pathway” -> “the genes in Wnt signaling pathway”

Line 42-44, "It is considered that the same direction of natural selection drives phenotypic changes among species since it is unlikely that these complex phenotypes have been acquired repeatedly just by neutral evolution". How about "Since it is unlikely that such a complex phenotype was acquired repeatedly by neutral evolution alone, the same direction of natural selection among species is likely to drive the parallel phenotypic change."?

We agree with your suggestion and correct the sentence of our manuscript.

Line 42-44: “It is considered that the same direction of natural selection drives phenotypic changes among species since it is unlikely that these complex phenotypes have been acquired repeatedly just by neutral evolution”

“Since it is unlikely that such a complex phenotype was acquired repeatedly by neutral evolution alone, the same direction of natural selection among species is likely to drive the parallel phenotypic change”

Line 60, polygenic  likely to be polygenic

We appreciate your comment. Indeed, it is better to weaken the wording.

Line 60: “most traits are polygenic” -> “most traits are likely to be polygenic”

Line 91, the Wnt  the genes in Wnt

We appreciate your correction. Last paragraph of introduction has been corrected according to the suggestion of Reviewer 2 (Q1).

Line 230, NovaSeq  Illumina NovaSeq

We appreciate your correction.

Line 222: “NovaSeq 6000” -> “Illumina NovaSeq 6000”

Line 231 "mRNA Library Prep Kit". Please add a company name.

We appreciate your correction. We added company’s information.

Line 223: “a TruSeq stranded mRNA Library Prep Kit.” -> “a TruSeq stranded mRNA Library Prep Kit (Illumina)”

Line 267, as for the tip of hypertrophied lips, could you add and point out which part is the tip?

We dissected hypertrophied lips in two half anterior and half posterior. We added the sentence in the materials and methods section.

Line 156-158: “The lips of *H. chilotes* were analyzed separately for the base and tip.” -> “The lips of *H. chilotes* were dissected in two half anterior (tip) and half posterior (base), which are analyzed separately.”

Line 272, "133 proteins upregulated and 5 proteins downregulated" in hypertrophied lip or normal lip?

We appreciate your correction. We added the sentence as follows.

Line 264: “133 proteins upregulated and 5 proteins downregulated”

“133 proteins upregulated and 5 proteins downregulated in the hypertrophied lip”

Line 274, "hypertrophied lips" means tip of hypertrophied lips?

We appreciate your correction. We corrected the sentence as follows.

Line 266: “hypertrophied lips are abundant” -> “tip of hypertrophied lips is abundant”

Line 277, Did you perform multiple testing correction for statistical significance?

We appreciate your comment about multiple testing corrections. We did not apply multiple testing corrections in our “exploratory” analysis of proteomics not to miss biologically important candidates in a limited sample size (n=3). We calculated the multiple corrected p-value in the Benjamini Hochberg method (Author response image 1, right). The result suggested that almost the same proteoglycans and its related proteins as we focused on are highly accumulated in the hypertrophied lips in milder conditions (significance level of 0.1).

**Author response image 1. sa2fig1:** 

Thus, our main conclusions remain unchanged even with correction applied, however, the overall balance of the volcano plot is not visually appealing (Author response image 1, right).

It is important to note that we selected the Top 20 proteins based on fold change rather than statistical significance. In addition, our proteomic findings show consistency with our histological and transcriptome data, providing the biological validation from various aspects. While we understand the potential benefits of multiple testing correction, our current approach without multiple testing still offers valuable and fair data to propose hypothesis on the molecular mechanisms of lip hypertrophy in cichlids. Therefore, we want to use original figure without multiple testing. We greatly appreciate the understanding of the reviewer.

Line 349-351, "The results of the enrichment analysis suggested that the genes that were categorized into both canonical and non-canonical Wnt signaling pathways, were highly expressed in the hypertrophied lips of juvenile and adult cichlids."The wnt category was enriched by analyzing the highly expressed genes, so isn't it natural that the wnt category is highly expressed?Did you mean to say as in the following sentence?"Enrichment of genes categorized in the canonical and noncanonical Wnt signaling pathways suggested that high expression of genes in the Wnt signaling pathway is likely to be involved in the hypertrophied lips of juvenile and adult fish."

Thank you for your comments. We corrected our manuscript as follows.

Line 341-344: “The results of the enrichment analysis suggested that the genes that were categorized into both canonical and non-canonical Wnt signaling pathways, were highly expressed in the hypertrophied lips of juvenile and adult cichlids.”

“As a result of enrichment analysis, DEGs were categorized in the canonical and noncanonical Wnt signaling pathways, suggesting that high expression of genes in the Wnt signaling pathway is likely to be involved in the hypertrophied lips of juvenile and adult fish.”

Line 403-404, "several other pathways may be involved in the development of hypertrophied lips". Do you have any evidence?

We appreciate your comment regarding possible evidence for the involvement of multiple pathways in hypertrophied lip development. Our statement was based on two main points:

(1) While we highlighted the Wnt pathway because this pathway is known to increase proteoglycan expression, we cannot exclude the possibility of the involvement of other pathways. For instance, our enrichment analysis in adult cichlids identified VEGF-related pathways, which could contribute to lip hypertrophy by increasing vascularization and nutrient supply to the lip tissue.

(2) Previous quantitative trait locus (QTL) analysis by Henning et al. (2017) concluded that lip hypertrophy is likely influenced by numerous loci with small additive effects. This indicates that lip hypertrophy is a complex phenotype consisted of multiple genetic factors, some which probably correspond to different molecular pathways.

Given these points, we draw a conclusion that emphasize the importance of Wnt pathway while also recognizing the potential cooperative interaction of multiple pathways in developing lip hypertrophy. Without confusing the two statements, we corrected our manuscript as follows.

Line 398-412: “We uncovered the apparent relationships between hypertrophied lips and the expression profiles of ECM proteins, in particularly proteoglycans. The trends for the overall expression of ECM-related genes were similar across hypertrophied lip species, but we rarely observed a specific gene that was commonly expressed at high or low levels in all three examples of hypertrophied lips across all East African Great Lakes. Furthermore, although we focused primarily on the relationship between the Wnt signaling pathway and lip hypertrophy, several other pathways may be involved in the development of hypertrophied lips. These findings imply that although enlargement of proteoglycan-rich loose connective tissue is common in hypertrophied lips, the developmental pathways to accomplish this are diverse in each lake.”

“We uncovered the apparent relationships between hypertrophied lips and the expression profiles of ECM proteins, in particularly proteoglycans. The trends for the overall expression of ECM-related genes were similar across hypertrophied lip species, but we rarely observed a specific gene that was commonly expressed at high or low levels in all three examples of hypertrophied lips across all East African Great Lakes. Furthermore, although we focused primarily on the relationship between the Wnt signaling pathway and lip hypertrophy, several other pathways may be involved in the development of hypertrophied lips. For example, our enrichment analysis in adult cichlids identified VEGF-related pathways, which could contribute to lip hypertrophy by increasing vascularization and nutrient supply to the lip tissue. In addition, previous quantitative trait locus (QTL) analysis by Henning et al. (2017) concluded that lip hypertrophy is likely influenced by numerous loci with small additive effects. These lines of data imply that although enlargement of proteoglycan-rich loose connective tissue is common in hypertrophied lips, the developmental pathways to accomplish this are diverse in each lake.”

**Reviewer 2:**
Minor comments:Last paragraph of Introduction: Remove the results of this study.

We appreciate your suggestion. We remove the specialized results from the last paragraph.

“In this study, we comprehensively compared the hypertrophied lips of cichlids across all East African Great Lakes using histology, proteomics, and transcriptomics. Histological and proteomic analyses revealed a distinct microstructure of hypertrophied lips compared to normal lips, and primary candidate proteins were identified. Transcriptome analysis at different developmental stages showed that the genes in Wnt signaling pathway was highly expressed in cichlids with hypertrophied lips at both the juvenile and adult stages. It is noteworthy that the distinct expression profiles observed in the proteome and transcriptome analyses of hypertrophied lips were similar among cichlids from each of the East African Great Lakes. The present study, which integrates comprehensive analyses for cichlids from all East African Great Lakes, provides insight for a better understanding of the molecular basis of a typical example of parallel evolution.”

Line 87-91: “In this study, we comprehensively compared the hypertrophied and normal lips of cichlids across all East African Great Lakes at various biological levels using histology, proteomics, and transcriptomics. As a result, we showed that a novel key pathway commonly involved in the formation of hypertrophied lips, providing insight into a better understanding of the molecular basis of a typical example of parallel evolution.”

Line 156: Italicize the scientific names.

We appreciate your correction.

Line 148: “M. zebra and O. niloticus” -> “*M. zebra* and *O. niloticus*”

Line 261: Remove the period after "Victoria."

We appreciate your correction.

Line 253: “Lake Victoria. (Figure 1; Figure S2).” -> “Lake Victoria (Figure 1; Figure S2).”

Line 416: Remove the period after "tissue."

We appreciate your correction.

Line 420: “tissue. (A,B)” -> “tissue (A,B)”

Line 646: Probably "the anterior side to the left."

We apologize for our mistake. As you commented, the anterior side is left. We corrected our manuscript as follows.

Line 648: “the anterior side to the right” -> “the anterior side to the left”

Fig. S2: Based on Fig. 1, the VG stained area appears larger in the Hypertrophied lip species; however, it is the opposite in Fig. S2.

We appreciate your comments. This is because we calculated the ratio of the VG-stained area to the whole lip area. While the absolute VG-stained area is larger in hypertrophied lips, the proportion of the VG-stained area relative to the total lip area is smaller. This correction using entire area allows us to simply compare the degree of lip hypertrophy among species.